# The Impact of Women’s Agency on Accessing and Using Maternal Healthcare Services: A Systematic Review and Meta-Analysis

**DOI:** 10.3390/ijerph20053966

**Published:** 2023-02-23

**Authors:** Maryam Vizheh, Frances Rapport, Jeffrey Braithwaite, Yvonne Zurynski

**Affiliations:** 1Australian Institute of Health Innovation, Faculty of Medicine, Health and Human Sciences, Macquarie University, Sydney, NSW 2109, Australia; 2National Health and Medical Research Council Partnership Centre for Health System Sustainability, Australian Institute of Health Innovation, Macquarie University, Sydney, NSW 2109, Australia

**Keywords:** maternal health services, health services, women’s agency, empowerment, systematic review, meta-analysis

## Abstract

Agency, defined as the ability to identify one’s goals and act upon them, has been recognized as a prominent strategy to access maternal healthcare services (MHS). The purpose of this study was to synthesize evidence of the association between women’s agency and MHS utilization. A systematic review was performed on five academic databases, comprising Scopus, PubMed, Web of Science, Embase, and ProQuest. Meta-analysis was performed with a random-effects method using the STATA™ Version 17 software. A total of 82 studies were selected following the PRISMA guidelines. The meta-analysis demonstrated that an increase in women’s agency was associated with a 34% increase in the odds of receiving skilled antenatal care (ANC) (OR = 1.34, 95% CI = 1.18–1.52); 7% increase in the odds of initiating the first ANC visit during the first trimester of pregnancy (OR = 1.07, 95% CI = 1.01–1.12); 20% increase in the odds of receiving at least one ANC visit (OR = 1.20, 95% CI = 1.04–1.4); 16% increase in the odds of receiving more than four ANC visits during pregnancy (OR = 1.16, 95% CI = 1.12–1.21); 17% increase in the odds of receiving more than eight ANC visits (OR = 1.17, 95% CI = 1.04–1.32); 13% increase in the odds of facility-based delivery (OR = 1.13, 95% CI = 1.09–1.17); 16% increase in the odds of using skilled birth attendants (OR = 1.16, 95% CI = 1.13–1.19); and 13% increase in the odds of receiving postnatal care (OR = 1.13, 95% CI = 1.08–1.19) compared to low level of agency. Any efforts to improve MHS utilization and reduce maternal morbidity and mortality should include the promotion of women’s agency.

## 1. Introduction

Maternal mortality continues to be a global health concern, particularly in low and middle-income countries [1]. According to the World Health Organization (WHO), in 2017, 810 women died every day worldwide, with a total of 295,000 deaths due to pregnancy and delivery-related complications that could have been prevented [1]. Moreover, for each maternal death, 20 to 30 women suffer from maternal morbidities worldwide—something on the order of 7.4 million events annualized [2]. 

Evidence demonstrates that almost 80% of maternal mortality is preventable if effective and high-quality maternal healthcare services (MHS) are provided throughout pregnancy, delivery, and during the postnatal period [3,4,5]. According to recommendations of the WHO, optimal MHS utilization is defined as the receipt of more than four antenatal care (ANC) visits during pregnancy provided by medically trained personnel (skilled ANC), with the first ANC visit during the first trimester of pregnancy; having facility-based delivery (FBD); having a skilled birth attendant (SBA) present during delivery; and receiving the first postnatal care (PNC) within the first 24 h of birth [6,7]. Since 2016, the WHO has recommended more than eight ANC visits during pregnancy because evidence has shown that fewer ANC visits are associated with a higher rate of perinatal death [6,8]. Moreover, receiving comprehensive and person-centered care at each visit is recommended to ensure the quality and maximum impact of MHS on the outcome of pregnancy [6]. The recommended adequate content of ANC visits includes routine screening for hypertensive diseases through regular monitoring of blood pressure; blood tests for anemia and HIV; tetanus vaccination; nutrition consultation; provision of iron/folate supplementation; vaginal examination; screening fetal heart rate; and at least two ultrasound examinations, including one during the first trimester [6,9].

The current literature on this subject, concentrates on potential factors which impede women from accessing adequate MHS. Some of the commonly reported barriers are: the non-availability of high-quality healthcare services, lack of knowledge of MHS and providers of these services, fear of stigma and judgment, long distances to health clinics, poor public transportation, lack of control over household resources/income, and cultural beliefs and practices which are more pronounced in developing countries [10,11]. However, strategies to encourage MHS receipt rely on the assumption that if high-quality services are provided, women will use them. Several studies, such as that of Finlayson and Downe (2013), suggest that this is not necessarily the case, as some women appear unlikely to access MHS, even when these services are available [12]. 

High rates of maternal mortality reflect not only health concerns but also social injustice, when governments and communities fail to safeguard women through a safe pregnancy, delivery, and motherhood experience [13]. High mortality of women during pregnancy and childbirth has been said to reflect a woman’s position in the household and more widely, in society, and in addition, an inability to access healthcare services and benefit from economic opportunities and social services [14]. Recently, a number of studies have attempted to understand how socio-cultural factors such as women’s position within the household or inequitable gender roles may impact the receipt of MHS [15,16]. These studies show that MHS is affected by women’s agency with the assumption that agency is fundamentally linked to women’s ability to make strategic life choices for themselves, such as accessing and using MHS [17,18,19]. 

Women’s agency is defined as the ability to identify one’s goals and act upon them [20]. Specific to access to and use of MHS, the agency has three key dimensions: (1) decision-making power, which promotes women’s ability to decide on their healthcare, household, and financial issues, which allows them to use these resources to achieve their goals including using MHS [20,21]; (2) freedom of movement, which is vital to cover the distance to health centers to access and receive needed MHS [21]. Women’s restricted and dependent mobility in traditional societies is also considered a pivotal obstacle to accessing MHS, as it suggests that women must be accompanied by another adult person, often their husband/partner or family-in-law, when travelling to access needed healthcare services [21,22]; and (3) the vocalization of gender-equitable attitudes, which improves women’s ability to have control over and access to financial resources to spend on their health, to have access to MHS, and to negotiate their reproductive choices with their partners [20,21]. One-quarter of women worldwide are disinclined to make decisions about the use healthcare services because of existing patriarchal and gender traditions which place them in compromising situations vis-à-vis their husbands/partners [23]. In turn, this limits their decision-making ability regarding access to and use of MHS [23]. 

A growing body of research has explored the association between women’s agency and use of MHS. However, the research reports mixed conclusions, with some studies reporting a positive association [24,25,26] and others finding no association between the two [27,28,29]. In addition, there are few systematic reviews on the relationship between women’s agency and MHS utilization. Two systematic reviews on the topic only partly address the impact of women’s agency on accessing and using MHS [30,31]. Pratley (2016), for example, examined the association between quantitative measures of women’s agency and maternal and child health outcomes, including receiving ANC and SBA in developing countries [31]. Mandal et al., 2017, on the other hand, analyzed the association between interventions to transform or accommodate prevailing gender norms and power imbalances with family planning and maternal health outcomes [30]. Both reviews showed that women’s agency is positively associated with increase in the odds of receiving MHS. They also highlighted several challenges, including the limited use of validated measures and the limited incorporation of dimensions of empowerment into measures (which in the case of these two articles is used as terminology synonymous with agency). They also reported difficulties in interpreting results because of the use of varied indicators and categorizations [30,31]. However, none of these reviews applied quantitative meta-analysis to synthesize the evidence. Moreover, both studies only considered a few number of MHS components in their analyses. Receiving ANC, PNC, and FBD and using SBA were the components of MHS included in these reviews [30,31]. 

It has been projected that between 2015 and 2030, almost four million women will die from maternal causes of death if the maternal mortality rate continues to decline at the present rate, which was a 2.9% decrease in maternal mortality rate annually between 2000 to 2010 [4]. The WHO has specifically stated that reducing maternal mortality requires guaranteeing universal access to MHS as well as eliminating inequalities and obstacles in accessing these services [32], yet it remains an unfinished agenda and a global challenge [33,34]. Agency may improve women’s access to MHS resulting in reducing maternal deaths; however, data on this association is not consistent. Understanding this relationship is particularly critical in settings with higher gender inequality and lower status of women that limit their agency to make strategic decisions on their health in accessing and using MHS [30,31]. 

To the best of our knowledge, no systematic review has quantified the association between women’s agency and uptake of MHS. Therefore, this study aimed to generate a synthesis of the existing evidence regarding the association between women’s agency and MHS utilization to inform health policies and advocacy efforts aimed to improve MHS utilization and reduce maternal mortality and morbidity. Thus, this review fills a critical empirical gap in global health by synthesizing and comparing quantitative evidence and quantifying the strength and direction of the association of women’s agency and access to and utilization of MHS. 

## 2. Methods

### 2.1. Study Design and Focused Question

This systematic review and meta-analysis was conducted according to the Preferred Reporting Items for Systematic Review and Meta-Analysis statement (PRISMA) [35]. The focused question was: “Does women’s agency can contribute to increasing their access to and utilization of MHS? And if so, what is the magnitude of this association?”. The focused question was developed using the Population, Exposure, Comparison, and Outcome (PECO) strategy [36]. This study investigated if women of reproductive age—15–49 years (Population)—with a high level of agency (Exposure) compared to women with a low level of agency (Comparison) have higher odds of accessing and using MHS (Outcome). 

Moreover, the research question was constructed in line with the FINER criteria [37]. These criteria investigated if the research question is Feasible (F), Interesting (I), Novel (N), Ethical (E), and Relevant (R) (Table 1). To formulate the research question, the authors investigated these criteria and rated each question as “yes” or “no” [38]. Regarding the research question in this study, the response for all components was “yes”. These criteria are presented in Table 1.

### 2.2. Literature Search Strategy

The search strategy was developed by two of the authors, MV and YZ. Five academic databases, comprising Scopus, PubMed, Web of Science, Embase, and ProQuest, were systematically searched. To capture additional, relevant articles that may not have been identified through searching the databases, a snowballing approach was applied to manually search reference lists of the included studies and study keywords and phrases accessed from Google Scholar. Various search strategies, such as keywords, index/subject terms, and medical subject heading (MeSH) terms were used to conduct the search. The exposure (women’s agency) and outcome (using and accessing MHS) were then combined using Boolean operators (AND/OR). For example, the Boolean operator ‘AND’ was used to link the primary concepts and terms used to express each concept were linked by ‘OR’ within brackets. An example of keywords used to search PubMed is presented in Table 2. 

### 2.3. Study Selection

Retrieved articles and abstracts were downloaded into EndNote V20, and duplicates were removed. Inclusion and exclusion criteria were applied when screening abstracts and reviewing full-text publications (Table 3). Screening of the titles and abstracts was performed by MV and YZ, and publications that met the inclusion criteria were subject to a full-text review using the selection criteria by MV and YZ, independently. Discrepancies were resolved by discussion, to come to a consensus of opinion. If consensus could not be reached, advice would be sought from the rest of the team (though this proved unnecessary). 

### 2.4. Data Extraction

A purpose-designed Excel spreadsheet, developed by the first author, was applied to extract relevant data from each article. The Excel spreadsheet was first piloted by MV and YZ on five articles, and further adjustments were applied as needed. The Excel spreadsheet was used to obtain relevant data on population characteristics (sample size, residence), study characteristics (study design, setting, data sources), women’s agency indicators, and MHS components. The associations between women’s agency and MHS were recorded for each study, and effect size (odds ratio (OR)) and statistical significance were extracted into the Excel file. OR was obtained directly from each study or calculated using the available data. In addition to the Excel spreadsheet, we used IBM Statistical Package for the Social Sciences (SPSS), version 27, to organize and analyze some of the numeric features of included studies, such as the number of associations between measures of women’s agency and MHS outcomes, number of studies conducted in each country and region, publication year, and sample size to achieve an overview of included studies.

### 2.5. Quality Assessment

To assess the risk of bias, the Mixed Methods Appraisal Tool (MMAT) was used. The MMAT is a critical appraisal tool validated for assessing the quality of studies of various designs and methodological approaches. In every category, there is a five-point checklist to be rated. This tool comprises the following domains in quantitative studies: sampling strategy, target population, measurements, risk of nonresponse bias, and statistical analysis [40]. MV and YZ assessed the quality of included studies independently. The accuracy of the ranking system was discussed and any difference between authors was resolved by discussion to reach consensus.

### 2.6. Data Analysis

The quantitative meta-analysis was conducted by MV in consultation with a statistician using STATA Corp. LLC (STATA), version 17. Given the heterogeneity found in the included studies, a random effects model (DerSimonian-Laird method) rather than a fixed effect model was applied to compute the pooled estimate of ln OR with a 95% confidence interval (95% CI) [41].

The assessment of heterogenicity was performed by MV and YZ, who also undertook the quality assessment of included studies. The degree of heterogeneity was expressed using Cochran’s Q test and I^2^ statistic [42], which was considered statistically significant at a *p* value of <0.05. Publication bias was evaluated through the visual inspection of the funnel graph and the calculation of Egger’s test [43]. 

In this review, the categorization of exposure (high and low agency) comes from the included studies. The included studies had scored the items used for measurement of agency. Then according to these scores, women who acquired higher scores on agency indices had been considered to have high agency and those who obtained low scores on these items had been considered to have low agency. High scores on indices of agency meant having greater decision-making power, freedom of movement, and acceptance of gender-equitable attitudes. Accordingly, using the information provided in the included studies, we extracted OR for high agency and OR for low agency for each individual study included in this review. Then, low agency was used as the reference category to compare the women with high agency and women with low agency regarding to accessing and using MHS. In addition, a subgroup analysis examined potential sources of heterogeneity by the dimensions of women’s agency and also by region where the study was conducted.

## 3. Results

### 3.1. Overview of Included Studies

Figure 1 displays the PRISMA flowchart [35] detailing the selection process of eligible articles for this review. The search of databases led to the retrieval of 8689 articles, leaving 6084 articles after excluding duplicates. The remaining 5811 articles were discarded after reviewing the titles and abstracts. Moreover, 43 records were retrieved through searching the references of included studies. Overall, 316 articles were subject to a full-text review. After reviewing full texts, 82 articles met all inclusion criteria for this review. 

Included studies were cross-sectional, with a sample size ranging from 137 women up to 765,169 women, from national Demographic Health Surveys (DHS). Most studies were carried out in sub-Saharan Africa and South Asia (68, 82.9%), and more than half the studies were conducted after 2018 (43, 52.4%), indicating that women’s agency is a recently emerging area of research which continues to evolve and expand. Secondary data using DHS, which is nationally representative data, was the source of data in 78% of included articles. In our systematic search, we did not find more than one study using a certain country’s DHS from a certain year. Notable characteristics of the 82 included studies are summarized in Table 4.

The quality appraisal by MMAT revealed that 27 out of the 82 included studies (32.9%) met all five quality criteria (Appendix A). Rejecting studies with low methodological quality is discouraged [40]. Nonetheless, we performed the quality assessment as an essential part of any systematic review to provide a more detailed presentation of the rating of each criterion to better inform the quality of the eligible studies in order to identify studies with high risk of bias. Then, if there was any study with poor methodological quality (high risk of bias), we would conduct a sensitivity analysis to explore whether the results might be affected if the studies at high risk of bias would be excluded [44]. However, none of the studies included in this review did not have high risk of bias, and therefore, no studies were excluded.

### 3.2. Dimensions and Domains of Indicators of Women’s Agency

In this study, 782 associations were found between various indicators of women’s agency and several MHS outcomes. Across the 82 included studies, 52 different items nested in 53 indicators of women’s agency were identified which were aggregated in diverse ways. Just 18 (21.9%) studies aggregated the indicators using exploratory factor analysis, confirmatory factor analysis, or principal component analysis (Table 5). Only two studies used validated scales, the Sexual Relationship Power Scale [45] and Survey-based Women’s emPowERment index (SWPER) [46]. One study used a scale comprised of 24 items without providing detailed descriptions of items [47]. 

The most used indicator of women’s agency was financial decision-making, followed by two aggregated indexes, gender-equitable attitudes, and one index involving three dimensions of healthcare decision-making, financial decision-making, and freedom of movement (used in 182, 159, and 135 associations, respectively).

### 3.3. Quantitative Meta-Analysis

Included studies used different quantitative methods to examine the association between women’s agency and MHS. Twelve studies either did not report a confidence interval (CI) or it was impossible to retrieve the CI from their reported data [22,48,49,50,51,52,53,54,55,56,57]. To maximize the number of included studies in the meta-analysis, we contacted the authors of these studies; however, we did not receive any further information. As a result, we excluded these studies from the quantitative meta-analysis. The remaining 70 studies were eligible for quantitative meta-analysis. 

### 3.4. Synthesis: The Associations between Indicators of Women’s Agency and MHS Outcomes

The most common outcomes identified across the studies were: using skilled ANC, timing of the first ANC, receiving at least one ANC visit, more than four ANC visits, and more than eight ANC visits, using SBA, FBD, and the receipt of PNC. The definitions of outcomes reported in the included studies are shown in Box 1. Few studies addressed some less-common outcomes, such as the quality of ANC and receipt of vaccination during pregnancy. As there were fewer than three studies that addressed such outcomes, meta-analysis was not possible. The outcomes that were associated with women’s agency and where there was enough data for analysis are in Box 1.

Box 1Definition of outcomes identified in the included studies.Skilled ANC: receiving any antenatal care from a skilled healthcare provider including a midwife, doctor, nurse, or from all three.Timing of the first ANC visit: receiving the first antenatal care during the first trimester of pregnancy.Receiving at least one ANC visit: receiving at least one antenatal visit during the most recent pregnancy.Receiving more than four ANC visits: receiving more than four antenatal visits during the most recent pregnancy.Receiving more than eight ANC visits: receiving more than eight antenatal visits during the most recent pregnancy.Facility-based delivery: giving birth at any health center, includ-ing a hospital, health clinic, or maternity clinic.Skill birth attendant: having a skilled healthcare provider for the most recent childbirth, including a midwife, doctor, or nurse, who has been trained in the skills needed to manage pregnancy and childbirth for the most recent childbirth.Receiving postnatal care: utilization of postnatal care by a skilled healthcare provider within the first 42 days after delivery for the most recent childbirth.

#### 3.4.1. Women’s Agency and Accessing Skilled ANC

Access to skilled ANC was evaluated in 31 studies (37.8%) through 107 associations. Receiving ANC from skilled providers showed the strongest association with greater agency. The pooled results of 26 studies eligible to be included in the meta-analysis revealed that an increase in women’s agency is associated with 34% increase in the odds of receiving skilled ANC (OR = 1.34, 95% CI = 1.18–1.52) (Appendix A). An Egger test, used to assess the publication bias, was insignificant (*p* = 0.74). However, the test for heterogeneity was significant (I^2^ = 94.14%, *p* < 0.001). 

#### 3.4.2. Women’s Agency and Initiating the First ANC Visit during the First Trimester of Pregnancy

This outcome was evaluated in 8 studies through 67 associations. Meta-analysis of 33 associations revealed that women’s agency is associated with a 7% increase in the odds of initiating the first ANC during the first trimester of pregnancy (OR = 1.07, 95% CI = 1.01–1.12) (Appendix A). While the test for heterogeneity was significant (I^2^ = 59.19%, *p* < 0.001), publication bias was not observed (*p* = 0.11). 

#### 3.4.3. Women’s Agency and Receiving at Least One ANC Visit

Receiving at least 1 ANC visit was examined in 10 studies through 66 associations. Quantitative meta-analysis on eight studies showed that an increase in women’s agency is associated with a 20% increase in the odds of receiving at least one ANC visit during pregnancy (OR = 1.20, 95% CI = 1.04–1.4) (Appendix A). An Egger test indicated no publication bias (*p* = 0.08). A significant heterogeneity was found among studies (I^2^ = 55.19%, *p* = 0.02).

#### 3.4.4. Women’s Agency and Receiving More Than Four ANC Visits

Receiving more than 4 ANC visits during the most recent pregnancy was examined in 33 studies (40.24%) out of 82 included studies, in which 166 associations were measured. The results of the meta-analysis on 26 eligible articles showed that women’s agency is associated with a 16% increase in the odds of receiving more than 4 ANC visits during pregnancy (OR = 1.16, 95% CI = 1.12–1.21) (Appendix A). Publication bias using Egger test was not seen (*p* = 0.39). Studies for this outcome indicated significant heterogeneity (I^2^ = 87.26%, *p* < 0.001). 

#### 3.4.5. Women’s Agency and Receiving More Than Eight ANC Visits

Three studies using 37 associations examined the association between women’s agency and receiving more than 8 ANC visits. The results of a meta-analysis on 23 associations showed that an increase of women’s agency is associated with a 17% increase in the odds of receiving more than 8 ANC visits during pregnancy (OR = 1.17, 95% CI = 1.04–1.32) (Appendix A). An Egger test did not show publication bias (*p* = 0.46). Studies showed significant heterogeneity (I^2^ = 78.50%, with *p* < 0.001). 

#### 3.4.6. Women’s Agency and FBD

Delivery in a health facility for the most recent pregnancy was the most common outcome of interest examined in 37 studies (45.12%) through 161 associations. Pooled meta-analysis conducted on 30 eligible studies revealed a significant association between women’s agency and FBD with a 13% increase in the odds of using these services compared to those with lower agency (OR = 1.13, 95% CI = 1.09–1.17) (Appendix A). Studies for this outcome indicated significant heterogeneity (I^2^ = 83.71%, with *p*-value of <0.001) and publication bias (*p* < 0.001).

#### 3.4.7. Women’s Agency and Having SBA

The association between women’s agency and SBA was examined in 31 studies (37.8%) through 107 associations, and of those, 26 studies involving 83 associations were included in quantitative meta-analysis. The results of pooled data revealed that an increase in women’s agency is associated with 16% increase in the odds of using SBA relative to those with lower agency (OR = 1.16, 95% CI = 1.13–1.19) (Appendix A). A significant level of heterogeneity (I^2^ = 74.68%) (*p* < 0.001) and risk of publication bias (*p* < 0.001) was seen among studies that assessed this outcome. 

#### 3.4.8. Women’s Agency and Received PNC

This outcome was examined in 16 studies (19.51%) through 54 associations of which 12 studies were included in the meta-analysis. Pooled analysis demonstrated a significant 12% increase in the odds of receiving PNC in women with the greater agency compared to those with lower agency (OR = 1.13, 95% CI = 1.08–1.19) (Appendix A). The test for heterogeneity was significant (I^2^ = 66.26%, *p* < 0.001). No significant level of publication bias was observed (*p* = 0.98). 

#### 3.4.9. Subgroup Analysis

Because there were insufficient studies for subgroups, we did not perform the subgroup analysis for studies that addressed receiving more than eight ANC visits. For other outcomes, subgroup analysis according to region showed that in South Asia, receiving more than four ANC visits is significantly associated with women’s agency, but this is not significantly associated in other regions (*p* < 0.001). Moreover, subgroup meta-analysis showed that initiating a first ANC during the first trimester of pregnancy is significantly associated with the decision-making dimension of agency (*p* < 0.001). Regarding the other outcomes, all subgroup analyses were insignificant (*p* ≥ 0.05).

## 4. Discussion

### 4.1. Summary of Evidence

This study, to the authors’ knowledge, is the first to systematically synthesize available evidence on the association between women’s agency and MHS utilization. Additionally, for the first time, this study applied quantitative meta-analysis. These analyses showed there are positive and significant associations between women’s agency and accessing and using MHS.

The uptake of MHS depends on both the demand for a service, indicating a perceived need to receive that service, and the recognition of the value and importance of available services, and women’s ability to act on service demand. Both gender relations and factors unrelated to gender relations impact these factors [49]. Women can face financial, geographic, and cultural constraints in terms of their ability to demand these services. On the one hand, physical proximity and improved access to these services can act as a resource which enables women to overcome external barriers to exercise agency [58,59]. For instance, in settings where women should be accompanied by a male guardian to cover the distance to go to these centers, the proximity of health centers to the household may waive the presence of a guardian to cover this short distance, resulting in greater ability to access health services. Moreover, short distances alleviate the cost of transportation, especially for women who face financial restrictions in receiving money from their husbands to spend on their health [59]. On the other hand, an improvement in women’s agency may also lead to greater access to such services [59]. Therefore, reducing these constraints appears to be a driver that would improve women’s access to MHS [60,61]. 

Expanding high-quality services and increasing women’s awareness of these services allows those women with agency to access them [62]. When MHS is not available, agency is irrelevant and neither women with high agency nor women with low agency can use them [62]. For example, in contexts in which geographic availability of healthcare services is limited, which is the case in many developing countries such as those included in this review, not having enough money or lack of knowledge of these services could be a barrier. In this case, women’s agency is not a main determinant. On the other hand, even if a woman has enough financial resources, knowledge, or both, her low level of agency can lead to low uptake of MHS [48].

In many traditional societies, social norms and patriarchy continue to significantly impact women’s access to MHS [63,64,65]. Sometimes, women’s decision-making power about whether to seek MHS is restricted and under the control of others in their families, such as husbands or family-in-law [60]. In these circumstances, any approach to improve women’s access to MHS must target cultural and social norms and beliefs [60].

### 4.2. Indicators and Indexes Used to Measure Women’s Agency

Included studies share a common feature methodologically in that they use similar or the same variables to represent women’s agency. Variables were potentially problematic as a result of the diversity in aggregation and operationalization of these indicators. Moreover, there was a variation in the included items within the composite indexes. While some studies aggregated items on various dimensions of agency in a single index, several studies used items as individual indicators of agency, limiting comparability across studies. 

Heterogeneity in indexes and measures of women’s agency is a constant issue also recognized in previous reviews [21,31,66]. One suggestion is to aggregate indicators using principal component analysis to reduce the number of dimensions and decrease the effect of multicollinearity, which could cause invalid statistical interferences [67]. It is has also been suggested that indicators should be operationalized and aggregated into meaningful dimensions [31]. Furthermore, Pratley (2017) highlights that the problems in measuring women’s agency may be rooted in the lack of a coherent conceptual framework for its measurement [31]. 

A further issue derived from this study was the lack of comprehensiveness and validity of indexes or items used to measure women’s agency. Most studies used a set of a few questions embedded in a DHS. A major advantage of the DHS system is that the same questions are asked across many women in many countries [65]. However, it has been criticized for being unable to capture the complexity and multidimensionality of women’s agency [68]. Agency is a multifaceted concept that varies between cultures and societies even within the same country [69]. Using DHS raised questions, as these indices may also point to geographical and contextual differences in agency. For instance, in societies where maternal health is considered a “woman’s domain”, women may seem to have more agency in accessing and using MHS than in societies where these issues is not regarded as a woman’s domain. Thus, due to specific cultural contexts of societies and differences in interpretation, these standard questions used in DHS to measure agency may not be equally valid across difference cultures [68]. Moreover, it remains uncertain whether and how these DHS questions have been validated. Others have also raised this concern and highlighted that the dimensions of agency used in these surveys are not internally consistent [68,70]. A full psychometric assessment such as a measurement invariance assessment and reliability tests is still needed to allow meaningful within-country, across-country, and longitudinal comparisons [71]. 

Despite calls in the previous literature to measure the broader structural dimensions of women’s agency [31,72,73], no further studies arose that assessed indicators of macro-level agency. This is particularly critical because women’s agency is largely dependent on how cultural beliefs and social norms impact its definition [74]. Entrenched social structures and culturally defined norms and practices that endorse gender inequality not only influence women’s access to the resource to exercise agency but also deprive them of making effective decisions and choices around their healthcare [75,76,77]. In aligning with previous studies, our review [31,72,73] further highlights the need for future research to develop conceptual frameworks and measures that consider women’s agency beyond an individual level.

## 5. Limitations

The results of this study should be considered in light of a number of limitations. Importantly, inconsistencies in how women’s agency is measured across the studies may have impacted our analysis. Although we employed the random effects model, aiming to control heterogeneity, the results should be interpreted carefully. Moreover, studies included in this review, mainly used population-based surveys such as DHS. While this method can make inferences across multiple countries, it misses the important nuance of context-specificity within and across these countries and may not capture the unique features of each context [78]. Another limitation was publication bias, as the inclusion criteria considered only peer-reviewed articles that reported quantitative data and excluded the grey literature, non-peer-reviewed reports, books, and dissertations as they were outside the scope of this review. Additionally, it includes only articles in English, which may have led to language bias. Future research should consider addressing these limitations.

Despite these limitations, these study findings provide an important contribution to the literature to date. Firstly, the study showed that women’s agency is significantly associated with increased odds of MHS utilization. A significant number of articles included in this study have helped deepen our knowledge of a wide range of settings where MHS and women’s agency is particularly pertinent. The study findings also emphasize the need to improve, validate, and standardize measures that can better capture the role of women’s agency in MHS utilization to support more robust comparisons and data aggregation across studies. Aggregated results could provide policymakers with an evidence-based platform to inform strategies for improving women’s agency and gender equality in their policies and programs to reduce maternal and child mortality and morbidity.

## 6. Conclusions

This review consolidates the notion that an increase in women’s agency is associated with increase in the odds of accessing and using MHS. It also brings to the fore the importance of developing and using comprehensive and validated indicators that measure clearly defined dimensions of women’s agency. There is a need for more in-depth disaggregated examination and appraisal of the measures of agency. The use of such indicators consistently across studies will enable greater aggregation of research evidence and comparison. Therefore, this study voices the need to improve the measurements of agency.

The positive association between women’s agency and MHS utilization suggests that improving women’s agency should be considered when developing policies and programs to make further progress in improving MHS utilization and maternal health. In the backdrop of the high incidence of maternal mortality and morbidity across the world, it is of paramount importance that improvement of women’s agency as an important sociocultural determinant of health be considered in health policies, strategies, and plans to improve women’s utilization of MHS and reduce the burden of women’s mortality. Women have the right to safe pregnancy, delivery, and motherhood. Achieving safe motherhood requires accelerating parallel efforts to expand MHS and to improve women’s agency to use MHS within the broader context of public health initiatives.

## Figures and Tables

**Figure 1 ijerph-20-03966-f001:**
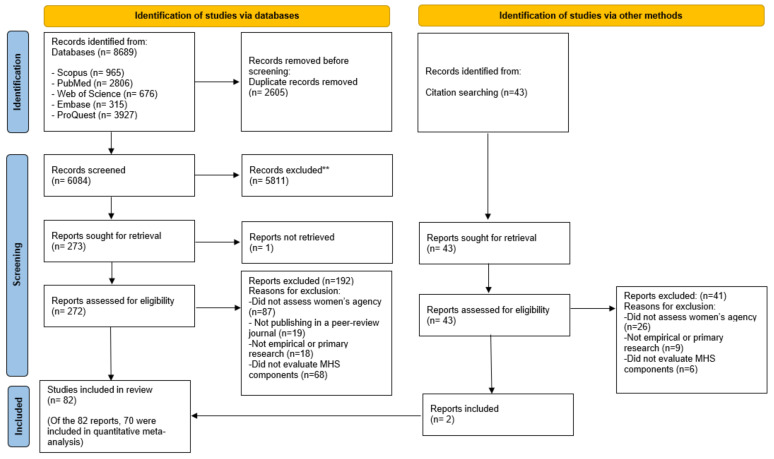
PRISMA flow diagram of searching and selection process.

**Table 1 ijerph-20-03966-t001:** FINER criteria for developing the research question.

Criterion	
Is the research question feasible?	Do authors have adequate time and funding? Do authors have an adequate number of subjects? Is the study manageable in scope? Is the study’s design appropriate?
Is it interesting?	Would other investigators, peers, and the community be interested in the study’s results?
Is it novel?	Does the study confirm or refute or extend previous findings? Does the study provide new findings?
Is it ethical?	Is the study ethical? Will the institutional review board approve it?
Is it relevant?	Is the study relevant to scientific knowledge, clinical and health policy, and future research?

Adapted from: Thabane et al., 2009 [37] and Gottlieb et al., 2018 [39].

**Table 2 ijerph-20-03966-t002:** Search strategy using Boolean operators.

Population	women OR female OR girl OR gender
	AND
Exposure	agency OR empowerment OR autonomy OR decision-making OR “healthcare decision making” OR “household decision making” OR “financial decision making” OR “freedom of movement” OR mobility OR “gender equality” OR “gender equity” OR “gender-based violence” OR status OR negotiation OR negotiating OR power OR coercion OR choice OR “bargaining power”
	AND
Outcome	pregnant OR pregnancy OR maternity OR mothers OR “preconception care” OR “maternal health service” OR “maternal care service” OR maternal OR “maternal health” OR “maternal care” OR “maternal healthcare utilisation” OR “prenatal care” OR “prenatal health” OR childbearing OR obstetrics OR perinatal OR “perinatal health” OR “perinatal care” OR “perinatal health services” OR “perinatal health outcomes” OR “perinatal health indicators” OR “perinatal health status” OR “antenatal care” OR ANC OR “ANC initiation” OR “quality of ANC” OR “antenatal care utilisation” OR “antepartum care” OR “antenatal period” OR “antenatal service” OR “antenatal care visit” OR “antenatal attendance” OR “professional delivery care” OR intrapartum OR birth OR parturition OR delivery OR “delivery care” OR “institutional birth” OR “delivery at health facility” OR “skilled birth attendant” OR “skilled delivery” OR “birth attendance” OR “childbirth” OR “postpartum” OR “postpartum care” OR “postpartum health” OR “postpartum period” OR “postnatal care” OR PNC OR “PNC visit” OR “reproductive health outcomes” OR “reproductive health indicators” OR “health services” OR “health care” OR “health services utilisation” OR “health access” OR “health outcomes” OR “health indicators” OR “health status” OR “quality care” OR “health care quality”

**Table 3 ijerph-20-03966-t003:** Inclusion/exclusion criteria.

Criterion	Inclusion	Exclusion	Rationale
Study design	Quantitative or mixed methods with a stand-alone quantitative portion	Qualitative or mixed methods without a stand-alone quantitative portion	This allows the comparability of findings across the studies
Study setting	Without limitation	N/A	Confining research to specific settings would restrict our knowledge of the available evidence
Study population	Women and girls	N/A	Women of reproductive age (15–49 years) who gave birth
Language	English	Non-English	The authors speak English
Document type	Peer reviewed	Non-peer reviewed; grey literature; posters; conference proceedings; editorials, opinions, theses and dissertations	Excluding grey literature and including peer-reviewed articles ensure high-quality and comparable research
Types of research	Primary empirical research or secondary analyses of data	Systematic reviews; protocols	This ensures the comparability of findings across studies and reasonable combining of the data
Exposure measure	Assessing at least one component of women’s agency, including decision-making power, freedom of movement, and gender-equitable attitudes	Articles measured other proxies of women’s agency, such as education, socioeconomic status, and demographic characteristics, without assessing any of the three components of women’s agency	Women’s agency may include three components: decision-making power, freedom of movement, and gender equitable attitudes
Outcome measure	Examining access to and use of at least one component of MHS according to WHO recommendations	Any healthcare service utilization outcomes except MHS	This systematic review explores the impact of women’s agency on accessing and using MHS
Study results	Without limitation	N/A	No restriction was imposed by type of results. Studies with negative, positive, or null associations between women’s agency and access to and use of MHS were included

**Table 4 ijerph-20-03966-t004:** Characteristics of included studies in this review.

Study Characteristics	
Study Settings	Number of Studies (Percentage)
Bangladesh	11 (13.4%)
Nepal	10 (12.2%)
Pakistan	8 (9.8%)
Nigeria	8 (9.8%)
Ethiopia	4 (4.9%)
Kenya	4 (4.9%)
India	4 (4.9%)
Cameroon	3 (3.7%)
Egypt	3 (3.7%)
Ghana	3 (3.7%)
Tajikistan	3 (3.7%)
Uganda	3 (3.7%)
Guatemala	1 (1.2%)
Indonesia	1 (1.2%)
Guinea	1 (1.2%)
Zambia	1 (1.2%)
Albania	1 (1.2%)
Senegal	1 (1.2%)
Multi-country	12 (14.6%)
Total	82 (100%)
**Regions of conducted studies**	
Sub-Saharan Africa	36 (43.9%)
South Asia	32 (39%)
Europe and Central Asia	4 (4.9%)
Middle East and North Africa	3 (3.7%)
East Asia and Pacific	1 (1.2%)
Latin America and the Caribbean	1 (1.2%)
Multi-region	5 (6.1%)
Total	82 (100%)
**Study design**	
Cross-sectional	82(100%)
**Data source**	
Primary data	18 (22%)
Secondary data (DHS) *	64 (78%)
**Publication year**	
≤1997	0 (0)
1998–2002	2 (2.4%)
2003–2007	3 (3.7%)
2008–2012	12 (14.6%)
2013–2017	22 (26.8%)
2018–2022	43 (52.4%)

* DHS: Demographic and Health Surveys.

**Table 5 ijerph-20-03966-t005:** Characteristics of scales used to measure women’s agency in included studies.

Measurement Characteristics of Included Studies	
Aggregated Measures or Not?	Number (Percentage)
Aggregated	61 (74.4%)
Not aggregated	21 (25.6%)
Method of aggregation	
Summative index	39 (47.6%)
Exploratory factor analysis	4 (4.9%)
Confirmatory factor analysis	2 (2.4%)
Principal component analysis	12 (14.6%)
Not stated	4 (4.9%)
Type of measures	
Single index	26 (31.7%)
Multiple indices/dimensions	32 (39%)
Mixed aggregated and separate indicators	3 (3.7%)
Not aggregated	21 (25.6%)

## Data Availability

The datasets generated and/or analyzed during the current study are not publicly available as all data are included in the manuscript body.

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
