# Peer review of "The Impact of Women’s Agency on Accessing and Using Maternal Healthcare Services: A Systematic Review and Meta-Analysis"

_ijerph, 2023, doi:10.3390/ijerph20053966_

Round 1
Reviewer 1 Report
1. The authors have synthesized the literature on women’s autonomy on use of healthcare services, which is very crucial to minimize maternal and child mortality especially in low- and middle-income countries.
2. Rationale of the study and objective was clearly stated.
3. The study is methodologically sound except related to statistics (Please ask statisticians to thoroughly check the statistical part).
However, it is not justifiable whether most of the studies drawn from DHS data can explicitly represent the indicators of women’s agency. For example; women’s agency may vary according to the context and community. Authors may justify these issues in discussion?
Reviewer 2 Report
Dear Authors
Thank you for the opportunity to read your manuscript, which I read with great interest.
The manuscript is well structured, however, it needs some changes that will improve it significantly. Below you will find some points in the manuscript that need clarification, refinement, reanalysis, rewriting, and/or additional information and suggestions on what can be done to improve it.
Title - Appropriate
Abstract - Some descriptors should be revised and brought in line with DeCS/Mesh: Maternal healthcare services - correct to Maternal Health Services; Healthcare services - correct to Health Services; Women's agency - not a descriptor; Women's Empowerment - correct to Empowerment
Section 1 (Introduction) - this section needs some adjustments, as some information and/or points are missing or unclear, and should be included or better written, I will present some items:
- What is the importance of doing this research/contribution it brings to the literature in the field?
- Why should readers be interested?
- What problem/issue does this research solve/fill?
- How will the proposed study address this deficiency/lacuna/problem and provide a unique contribution to the literature.
- Objective of the review, which should be in line with that presented in the abstract.
Section 2 (Materials and Methods) - in this section some points should be clarified and improved and included, namely:
- What is the research question, which should be constructed in line with the objective and with Strategies for assessing the FINER criteria (FINER: Feasible, interesting, novel, ethical, and relevant) and PICO format, I suggest reading the article https://link.springer.com/article/10.1007/s12630-008-9007-4
- Clarifying the Boolean search equation.
- Clarification of the search criteria, namely the type of participants and the results
- Could have presented more information about the data extraction table, as unclear text, did you use Excel and SPSS?
- It seems to me that it makes more sense for point 2.5. Quality evaluation, before the point 2.4. data analysis.
- In point 2.5. Quality Assessment, the Mixed Methods Appraisal Tool (MMAT) should be clarified.
Section 3 (Results) - This section leaves me in doubt, because not having some missing methodological information mentioned earlier, I cannot clearly evaluate the data.
- In this section, they present the quality assessment using the MMAT, in which only 27 studies had a good methodological quality and then state that they consider all of them, I do not understand why they then performed the methodological assessment if they did not consider the results? It needs better substantiation.
- Figures and supplementary tables are difficult to read due to the quality presented.
Section 4 (Discussion) - the discussion leaves me in doubt, as I don't know the methodological issues specifically, I can't make a correct assessment, however, it seems reasonable
Section 5 (Conclusion) - I could present strategies to reduce the problems in study.
Reviewer 3 Report
GENERAL COMMENTS:
The paper is a systematic review with metanalysis of the association between womens’ agency and access to maternal health services.
The main contributions of the paper are: (i) synthesizing and comparing quantitative evidence; (ii) quantifying the effect of womens’ agency on access to maternal health services.
The main strengths of the paper are: (i) robust study methodology and execution, providing relevant evidence; (ii) providing a quantitative measure of evidence to advocate for increasing women’s agency as part of the social determinants of health.
SPECIFIC COMMENTS:
Abstract
Brief definitions should be provided for “women’s agency”, along with the scale used to quantified it.
In line with WHO global recommendations, it would be better to look at the following indicators for adequate number and timeliness of ANC visits: first ANC visit during the first trimester; at least one ANC visit during pregnancy (rather than more than one visit), more than four ANC visits during pregnancy, and more than eight ANC visits during pregnancy. Furthermore, it would be better to also a relevant indicator for adequate content of ANC visits, for instance: at least two ultrasound tests including one during the first trimester.
Introduction
Lines 55-57: the Finlayson and Downe study may be referenced only once, rather than twice.
Line 69: “Women’s agency is defined as the ability to (…)”
Lines 78-80: reference?
Lines 98-99: Which ones were used? Which ones could but were not used, according to the authors?
Methods
Lines 110-111, 128-130, 136-138, 148-149, 153, 165-166: the authors role should not be in the main text, but rather in the dedicated Authors’ contributions section.
Lines 107: all the keywords search strings should be provided in full, not just “for example”.
Lines 157-158: The scale used to quantify “A low level of (women’s) agency (…)” should be provided.
Results
Lines 231-233: do you mean “more than one”, “more than four”, and “more than eight” ANC visits?
Lines 234-235 and 300-301: should go in the Limitations section.
Box 1 and line 254: In line with WHO global recommendations, it would be better to formulate the first indicators as “receiving at least one ANC visit” (rather than more than one visit).
Discussion
Line 310: better to add caveat (e.g., “to the authors’ knowledge).
Lines 341-352: content should be captured in the Limitations section.
References
#1-2 and 4 are not suitable and should be replaced with WHO references.
#6 should be replaced with a key reference on the association between access to high-quality maternal health services and maternal mortality.
#8 and 10 should be replaced by the WHO reference.
#11 should be supplemented with another key reference on access barriers to ANC, in addition to obstetric care.
#12 should be replaced with a reference with global relevance, rather than about Nepal only.
#14 should be replaced with a reference with global relevance, rather than about Nigeria only. Alternatively, additional and diverse country-specific references may be added to support the statement.
#15- 16, 7 and 9, 20, 37, 38, 48, and 50: should be supplemented with additional and diverse country-specific references to support the statement.
#21-22, e.g.23: what about #20 (seems to be missed). Also, no need to write “e.g.,” in reference annotation.
e.g.26: No need to write “e.g.,” in reference annotation.
Reviewer 4 Report
Thank you for the opportunity to review this interesting paper, on an important topic. I have a few comments, however,
I am struck by this statement, “Both reviews suggested improving women’s agency as a viable strategy to improve maternal and child health.” There is a causality implied (that more agency means more access to MHS), that could be simply two correlated variables. And improving women’s agency is being talked about as if it were a widget to simply change women’s status in their families/communities.
Methods: it is stated, “A low level of agency was used as the reference category.”, but not how this was measured or determined.
Results: most of the studies included used DHS data. Did the review include multiple studies using the same dataset? For example, if more than one study used a certain country’s DHS from a certain year, were all of those studies included in the meta-analysis?
The authors are reporting the odds ratios as risk ratios (“an increase in women’s agency is associated with 34% increase in receiving skilled ANC (OR=1.34, 95% CI=1.18-1.52)”. An odds ratio is not a risk ratio, and there is a 34% change in the odds of receiving. I suggest the authors use marginal effects (Stata command: margins) to determine the % increase in the likelihood of any outcome.
Discussion: there are causal arguments being made which are not supported by these cross-sectional data. For example, “It showed that women who possess higher agency are significantly more likely to use MHS.” This is not quite true. These analyses showed there is an association between using MHS and these measures of agency.
The authors write, “On the one hand, physical proximity and improved access to these services can stimulate agency.” It is not clear to me how improved access, either geographic or otherwise, can increase a woman’s agency.
They next write, “Expanding high-quality services and increasing women’s awareness of these services may increase women’s agency to use them [49]. When MHS is not available, neither women with high agency nor women with low agency could use them [49].” I think the second sentence here is correct, but not the first; if the services are not available, agency is irrelevant. Only when they are available does a woman’s agency matter. So, having services available does not “increase women’s agency”, but rather allows those with agency to access them.
The authors write, “Most settings in the included articles are characterized by social norms that perpetuate the status of women as subordinate and subservient to men.” This is quite a statement to make without any reference or explanation.
The limitations section appears contradictory. While it is stated, “Importantly, inconsistencies in how women’s agency is measured across the studies, which may have impacted our analysis. Although we employed the random effects model, aiming to control heterogeneity, the results should be interpreted carefully.” However, this is followed by, “the study provides a robust set of estimates for the role of women’s agency in accessing and using MHS.” I agree with the authors that measuring something as complex as agency, which is a culturally dependent construct, across settings, with a single (or maybe 3-question) actualization is complicated and these analyses may well oversimplify this broad and multifaceted idea. I am not convinced by these analyses that the measures of agency in the DHS and the outcomes aren’t both measuring something else. I suggest that families or communities in which women have freedom of movement are likely more inclined to access health services for those who need them than places or families in which this is not the case.
Conclusion: I am not clear on what the authors mean by, “effective approaches to improve women’s agency”. It would be helpful to have some examples, as the suggestion that a government or ministry of health can improve women’s agency seems far-fetched to me.
Round 2
Reviewer 2 Report
Dear Authors
Thank you for your attention to my comments and suggestions, however in Section 2 (Materials and Methods) - in this section some points should be clarified and improved and included, namely:
- What is the research question, which should be constructed in line with the objective and Strategies for assessing the FINER criteria (FINER: Feasible, interesting, novel, ethical, and relevant) and PICO format, I suggest reading the article https://link.springer.com/article/10.1007/s12630-008-9007-4
- In point 2.2, the presentation of the Boolean equation for each base of the dos, could have a clearer presentation, namely in the table presentation.
- Figure 1 PRISMA should meet the models presented https://www.prisma-statement.org//PRISMAStatement/FlowDiagram
